# Ovary of Zebrafish during Spawning Season: Ultrastructure and Immunohistochemical Profiles of Sox9 and Myostatin

**DOI:** 10.3390/ani13213362

**Published:** 2023-10-29

**Authors:** Dalia Mohamedien, Doaa M. Mokhtar, Nada Abdellah, Mahmoud Awad, Marco Albano, Ramy K. A. Sayed

**Affiliations:** 1Department of Histology, Faculty of Veterinary Medicine, South Valley University, Qena 83523, Egypt; dalia.mohamadain@vet.svu.edu.eg (D.M.); mahmoud.awad@vet.svu.edu.eg (M.A.); 2Department of Cell and Tissues, Faculty of Veterinary Medicine, Assiut University, Assiut 71526, Egypt; doaa@aun.edu.eg; 3Department of Histology and Anatomy, School of Veterinary Medicine, Badr University in Assiut, New Nasser City, Assiut 11829, Egypt; 4Department of Histology, Faculty of Veterinary Medicine, Sohag University, Sohag 82524, Egypt; nada_abdalah@vet.sohag.edu.eg; 5Department of Veterinary Sciences, University of Messina, Polo Universitario Dell’Annunziata, 98168 Messina, Italy; 6Department of Anatomy and Embryology, Faculty of Veterinary Medicine, Sohag University, Sohag 82524, Egypt; ramy.kamal@vet.sohag.edu.eg

**Keywords:** theca cells, ovarian follicles, electron microscopy, oogenesis, ovarian stroma

## Abstract

**Simple Summary:**

Reproduction is a pivotal physiological process in various biological organisms. As the zebrafish has been suggested as a model for the study of numerous aspects of ovarian development, this research aimed to pinpoint the cellular and microenvironmental changes of the ovary in zebrafish during the spawning season, in addition to providing information on the ovarian expression of Sox9 and myostatin. Various stages of oogenesis were described. Furthermore, immunohistochemical analysis revealed SOX9 and myostatin expression in primary and pre-vitellogenic oocytes and the theca cell layers. Moreover, a wide diversity of cells was detected in the ovarian stroma. Collectively, the findings of this study support the importance of the zebrafish ovary as a model for follicular development studies.

**Abstract:**

This study sought to examine the ovarian cellular and stromal components of the zebrafish (*Danio rerio*) throughout the spawning season using light and electron microscopic tools. The ovaries of zebrafish showed oocytes in all stages of follicular development and degeneration (atresia). Six stages of oogenesis were demonstrated: oogonia, early oocytes, late oocytes, vacuolated follicles, the yolk globule stage (vitellogenesis), and mature follicles. The SOX9 protein was expressed in the ooplasm of the primary and previtellogenic oocytes and the theca cell layer of the mature follicles. Myostatin was expressed in the granulosa and theca cells. Many stem cells in the ovarian stroma expressed myostatin and SOX9. During the spawning season, the EM results indicated that the zona radiata increased in thickness and was crossed perpendicularly by pore canals that contained processes from both oocytes and zona granulosa. The granulosa cells contained many mitochondria, rER, sER, and vesicles. Meanwhile, the thecal layer consisted of fibroblast-like cells. Atretic follicles could be demonstrated that involved both oocytes and their follicular walls. Several types of cells were distinguished in the ovarian stroma, including mast cells, telocytes, lymphocytes, fibroblasts, endocrine cells, macrophages, adipocytes, dendritic cells, and steroidogenic (stromal) cells. The ovary of the zebrafish serves as a model to investigate follicular development.

## 1. Introduction

Reproduction is a pivotal physiological process in various biological organisms. Multicellular systems use gametes, which emanate from the primordial germ cells during embryogenesis for their reproduction. In the vertebrates’ ovary, each follicle encompasses three major cellular elements: oocytes, inner granulosa cells, and outer theca cells. The ovarian follicles of fish display a single layer of granulosa and theca cells, in contrast to mammalian ovarian follicles, which are composed of numerous layers of follicular cells [1].

The zebrafish (*Danio rerio*) has become a prestigious model for the study of vertebrate development and physiology over the past few decades [2]. Numerous aspects of ovarian development may be studied using the zebrafish as a model [3,4]. The fully sequenced genome, tiny size, resistance, and ease of genome modification of the zebrafish are its most favorable traits. The zebrafish may also be made to breed artificially or naturally all year long in a lab setting [5]. They have an unusual and extensive spawning strategy, where the fish prefer to spawn in social environments. Moreover, zebrafish are known to spawn multiple times over the course of several days. The female releases her eggs after fertilization, which float in the water in the form of clusters [6]. The reproductive cycle of the zebrafish is not very long. Depending on environmental factors such as temperature, the eggs hatch in approximately 48–72 h post-fertilization [7]; thus, they are a significant model organism for scientific studies because of their short reproductive cycle [8]. The age and size of the female, the habitat, and genetics are some of the variables that can affect the number of spawned eggs per batch and the zebrafish fecundity estimate. When zebrafish spawn, a single female can deposit a significant quantity of eggs. It is believed that zebrafish have a high fecundity rate [9,10]. Over the course of several spawning events over a week, a single female can generate several hundred to over a thousand eggs [11].

One of the crucial transcription factors in the development of various tissues and organs, including sex determination and chondrogenesis, is the SRY-box-containing gene 9 (SOX9 gene). SOX9 is a member of the SOX gene family. Up to this point, fish, birds, reptiles, amphibians, nematodes, and insects have all been found to include members of this family [12]. In the testes and ovaries, SOX9 is upregulated and downregulated, respectively, during gonad differentiation. The significance of SOX9 in the male developmental route is suggested by this expression pattern and the male sex reversal observed in SOX9-deficient individuals, although its role in sex differentiation is yet unknown [13]. Two SOX9 genes, SOX9a and SOX9b, have been found in zebrafish. Both are expressed in chondrogenic tissues. SOX9b is localized in the ovary, while SOX9a is expressed in the testis of the zebrafish gonad. These findings indicate that whereas SOX9 expression in the gonad appears to be slightly diverse in fish species, it is conserved in the chondrogenic tissues of vertebrates [13].

Myostatin or growth differentiation factor-8 (GDF-8) is a protein that belongs to the superfamily of transforming growth factor β (TGF-β) and was first identified as a negative regulator of skeletal muscle growth [14]. Myostatin has been implicated in the regulation of adipocyte and cardiomyocyte function, in addition to the skeletal muscles [15]. Notably, recent studies have examined the expression of myostatin and its putative functions in several reproductive tissues, such as the uterus, ovary, and placenta. The modulation of follicular growth, granule cell proliferation, ovarian steroidogenesis, and regulation of oocyte maturation proposed to be enabled by myostatin show how important myostatin is for human reproduction and fertility [16]. Therefore, this study aimed to provide details on the ovarian expression and potential intraovarian function of SOX9 and myostatin. Concurrently, the cellular and microenvironmental changes in the ovary in zebrafish during the spawning season were investigated.

## 2. Materials and Methods

This study was performed according to Egyptian laws and university guidelines for animal care. All procedures were approved by the National Ethical Committee of the Faculty of Veterinary Medicine, Sohag University, Egypt, under Ethical No. Soh.aun.vet/00040.

### 2.1. Sample Collection

Healthy mature adult female specimens of zebrafish (n = 8), aged 4–5 months, that were randomly selected and measured 2.5 ± 0.5 cm in standard length were used in this study. As previously reported, zebrafish reach sexual maturity at the age of 3–4 months, with the standard length reaching 20–30 mm [17,18]. The fish were collected in November 2021 from the Laboratory of Developmental Biology, Graduate School of Integrated Science of Life, Hiroshima University, Japan, where the fish were monitored from the time of hatching till reaching the age of 4–5 months. Following anesthesia using isoflurane (095-06573, FUJIFILM Wako Pure Chemical Corporation, Osaka, Japan), a longitudinal incision was made on the abdominal wall of the fish, and the ovaries were dissected and fixed for further examination.

### 2.2. Histological Analysis

The ovaries of zebrafish were immediately dissected and were directly immersed in Bouin’s fixative fluid for 22 h. After proper fixation, the ovaries were processed for histological examination. Briefly, the specimens were dehydrated with ethanol and cleared in 99% methyl benzoate, and, finally, they were embedded in paraffin wax. Then, 5-μm-thick transverse sections were obtained and stained with Sirus red (Product No. 365548, Sigma-Aldrich, Madrid, Spain) [19].

### 2.3. Immunohistochemical Detection

Immunohistochemistry was carried out on paraffin sections of zebrafish ovary (5 μm thick) using the Pierce Peroxidase Detection Kit (36000, Thermo Fisher Scientific, Waltham, MA, USA). Briefly, the sections were dewaxed with xylene, rehydrated in ascending grades of ethanol, and washed with distilled water. The sections were heated in a sodium citrate buffer (0.01 M, pH 6.0) for 15 min to increase epitope exposure. After cooling at room temperature for 30 min, the sections were then washed with wash buffer (Tris-buffered saline “TBS” with 0.05% Tween-20 Detergent). To suppress the endogenous peroxidase activity, sections were then incubated in a peroxidase inhibitor for 30 min. Following this, the sections were blocked using a Universal Blocker^TM^ blocking buffer (36000, Thermo Fisher Scientific, Waltham, MA, USA) in TBS after being washed with a wash buffer. Overnight incubation (4 °C) of the tissue sections was performed with the primary antibodies (1:100) against SRY-Box transcription factor 9 (SOX9) (AB5535, Sigma-Aldrich, Madrid, Spain) or myostatin (AB3239, Sigma-Aldrich, Madrid, Spain). Ovarian tissue, in which the SOX9 primary antibody was omitted and replaced with a buffer, served as a negative control (Appendix A). After incubation with primary antibodies, sections were incubated with goat anti-rabbit IgG (65-6140, Invitrogen, Waltham, MA, USA) secondary antibodies at a dilution of 1:1000 at room temperature for 30 min after being washed with wash buffer. The tissues were then treated for 30 min with diluted (1:500) Avidin-HRP (43-4423, Invitrogen, Waltham, MA, USA) in Universal Blocker blocking solution after the sections had been rinsed with a wash buffer. After this, the tissues were rinsed again with a wash buffer, and the desired staining was achieved by incubating them with 1X metal-enhanced DAB substrate working solution (36000, Thermo Fisher Scientific, Waltham, MA, USA). Following this, the sections were washed with a wash buffer again and were counterstained with Harris-modified Hematoxylin (36000, Thermo Fisher Scientific, Waltham, MA, USA) before applying the mounting media (36000, Thermo Fisher Scientific, Waltham, MA, USA).

### 2.4. Semithin Sections and Transmission Electron Microscopy (TEM)

Small specimens from the ovaries were fixed overnight in a solution of 2.5% paraformaldehyde–glutaraldehyde [20]. After fixing, the samples were washed in phosphate buffer (0.1 Mol/L) and osmicated with 1% osmium tetroxide in sodium–cacodylate buffer (0.1 Mol/L, pH 7.3). Dehydration was performed with ethanol and propylene oxide, followed by Araldite. Toluidine blue was used to stain semithin sections that were 1 μm thick for evaluation under a light microscope. Ultrotom-VRV (LKB, Bromma, Sveden) was also used to cut ultrathin sections (70 nm thick), which were then stained with uranyl acetate and lead citrate [21]. An electron microscope, model JEOL-100CX II, was used to capture TEM images.

### 2.5. Digitally Colored TEM Images

Digital coloring was performed on the TEM images to enhance the visual contrast between different structures. Various elements within the same electron micrograph were carefully hand-colored using the Adobe Photoshop software version 6.

### 2.6. Morphometrical Analysis

Morphometrical analysis including the number and diameter (µm) of the different stages of developing follicles in the zebrafish ovary was carried out using images of Sirus-red-stained paraffin sections (3 images, 100× objective, per animal). Using the Image J (v. 1.52t) processing software, the measurements were examined by two blinded researchers and stated as mean ± *SEM*.

## 3. Results

### 3.1. Histological Analysis

#### 3.1.1. Follicular Development

The ovaries of zebrafish showed oocytes in all stages of follicular development and degeneration (atresia) (Figure 1A). The oogenesis included oogonia proliferation by mitosis and oocyte development. Six stages were observed in zebrafish based on morphological changes in the size, ooplasm, nucleus, and egg membranes of the developing follicles.

*Stage 1—Oogonia:* They were gathered in groups or nests under the germinal epithelium, and they were the smallest cells of the germinative lineage. They were small, spherical cells with a large nucleus. A thin film of yellow-stained ooplasm surrounding the nucleus was observed by Sirus red staining. All ovaries of developing and mature females had several patches of oogonia (Figure 1A). Oogonia were divided by mitosis to give the early oocytes. During the spawning season, their number reached 10 ± 1, and their diameter was 41.26 ± 10.9 µm.

*Stage 2—Early Oocytes (Chromatin Nucleolus Stage):* These oocytes were found neighboring oogonia (Figure 1A). The transformation of oogonia to oocytes involved an expansion in the size of the cell and nucleus. The number of nucleoli increased, they were dispersed across the nucleoplasm, and the ooplasm was reduced and deeply basophilic (Figure 2A–C). Their number and diameter were 11.6 ± 0.57 and 67.3 ± 6.62 µm, respectively.

*Stage 3—Late Oocytes (Perinucleolar Stage):* The oocytes increased progressively in diameter to 111.7 ± 3.9 µm and their number was 4.6 ± 0.57 (Figure 1B). The number of nucleoli increased, and they were located in the peripheral part of the nucleus. Oocytes were surrounded by a layer of squamous follicular epithelium (Figure 2B,C).

*Stage 4—Vacuolated Follicles (Yolk Vesicle or Cortical Alveolar Stage):* The nucleoplasm attained red granules with Sirus red stain (Figure 1B) and the nucleus increased in size to become the germinal vesicle (Figure 2A). Perinucleolar and yolk vesicle oocytes displayed many nucleoli that decreased in the yolk globule and mature oocytes. The ooplasm contained many small yolk vesicles, which were peripherally arranged, later became cortical alveoli, and took part in the formation of the perivitelline space. A few small cytoplasmic yolk granules could be detected around the nucleus. Zona radiata (oolemma) began to appear, which had the form an acellular thin hyaline membrane. The follicular layer was composed of cuboidal cells (Figure 2A). The oocytes increased both in number and in diameter. Their number and diameter reached 2 ± 0.8 and 149.3 ± 15.2 µm, respectively.

*Stage 5—Yolk Globule Stage (Vitellogenesis):* The oocytes increased in diameter to 271.4 ± 35 µm, and their number was 1.8 ± 0.8. In addition, the yolk vesicles increased in size in this stage. The yolk granules (protein) coalesced into large yolk globules. Numerous rounded yolk globules (platelets) could be observed near the center of the oocyte and extended centrifugally until only a thin peripheral shell of cytoplasm remained (Figure 1C). Numerous fat vacuoles were distributed throughout the ooplasm. The cortical alveoli were represented by small vesicles located close to the oocyte periphery. The zona radiata and the follicular epithelium (granulosa) were thick (Figure 2D).

*Stage 6—Mature Follicles:* Mature follicles were characterized by the presence of many empty large vacuoles toward the oocyte periphery and the yolk globules increased in size. The nucleus (germinal vesicle) gradually disappeared, which usually occurs at the end of vitellogenesis (Figure 1D). The yolk globules consisted mainly of lipoprotein with a small proportion of carbohydrates and displayed strong reactivity to the Sirus red stain (Figure 1D). The protein yolk globules coalesced as the germinal vesicle broke down, and the oocytes rapidly increased in diameter to reach 444.5 ± 59.5 µm; then, the oocyte ovulated into the ovarian lumen and became a mature ovum. Upon completion of this phase, the nucleus was not visible in the maturation stage due to the disintegration of the nuclear membrane and dispersion of its content in the cytoplasm. In addition, the follicular layers became extremely well developed, consisting of zona radiata that attained their maximal thickness and were surrounded by cuboidal granulosa and theca cells. An outer layer of collagenous connective tissue and an inner layer of theca cells made up the theca folliculi (Figure 2E,F). The number of mature oocytes reached 3 ± 2.6.

#### 3.1.2. Atretic Follicles

Atretic follicles result from the degeneration of an oocyte and its resorption by phagocytosis. The atretic follicular cells enlarged and phagocytized the oocytes, which did not spawn, and the connective tissue surrounding the follicle thickened. The follicles were contracted and folded (Figure 3A). Their diameters were too variable to be measured since they were irregular structures that could be demonstrated at any time during the maturation of the oocyte. The degradation and regression of yolk granules in the peripheral ooplasm with the hypertrophy of follicle cells were demonstrated. In the last stage of atresia, the ooplasm contained vacuoles and numerous small masses of unclear nature that gave a structure that resembled the corpora atretica. Moreover, the follicles decreased in size and the yolk was completely reabsorbed, while the zona radiata and follicular cells were completely digested (Figure 3B).

During spawning, the stroma was extremely pressed due to the enlargement of the ova and the ovarian lamellae were thick and completely obliterated the ovaries (Figure 1C,D and Figure 2A,B). The stroma contained various types of cells, including fibroblasts (Figure 3C,D), telocytes, mast cells (Figure 3E), fat cells, stromal cells, macrophages, and other immune cells (Figure 3F).

### 3.2. Immunohistochemistry

The SOX9 protein was expressed in the ooplasm of primary and pre-vitellogenic oocytes (Figure 4A,B) and the theca cell layer, which surrounded the mature follicles (Figure 4C,D). Many stem cells in the ovarian stroma expressed SOX9 (Figure 4E,F). Myostatin was expressed in theca and granulosa cells (Figure 5A–C). The stem cells in the ovarian stroma showed immunoreactivity to myostatin (Figure 5D).

### 3.3. Transmission Electron Microscopy (TEM)

In the outer ooplasm of the mature follicles, lipid droplets and yolk globules could be distinguished (Figure 6A). The zona radiata appeared electron-dense and compact (Figure 6A,B). With the advancement of development, the zona radiata increased in thickness and was crossed perpendicularly by pore canals that contained processes from oocytes and zona granulosa (Figure 6C). Two cell layers, an outer thecal layer, and an inner granulosa layer, were present in the mature ovarian follicles and were separated by a basement membrane (Figure 6D). During development, the granulosa layer changed from squamous to cuboidal cells. The cytoplasm of these cells contained many mitochondria, rER, sER, and vesicles. The thecal layer consisted of fibroblast-like cells, blood capillaries, and some collagen fibers (Figure 6D). In addition, telocytes (TCs) with a cell body comprising a spindle euchromatic nucleus and cell processes (telopodes) could be distinguished in the thecal layer. The telopodes contained many secretory vesicles (Figure 6C). Macrophages with heterogeneous cytoplasmic content (Figure 6D) and lymphocytes, which had a high nuclear to cytoplasmic ratio and heterochromatic nucleus, were seen in the ovarian stroma (Figure 6C).

The stroma of the ovary contained many steroid interstitial stromal cells that were distinguished by the presence of sER, mitochondria, and many lipid droplets (Figure 7A). Dendritic cells could also be seen in the ovarian stroma in association with the blood vessels or near macrophages. These cells displayed fine dendritic processes and indented nuclei (Figure 7B,C). Monocytes with a kidney-shaped eccentric nucleus and cytoplasm displaying many vacuoles were observed in close contact with blood vessels (Figure 7D). Stem cells were noticed in the stroma that showed dividing nuclei (Figure 7C) and a high nuclear to cytoplasmic ratio (Figure 7D).

Furthermore, the ovarian stroma of the zebrafish contained fat cells that were characterized by the presence of many fat droplets in their cytoplasm (Figure 8A,B). Telocytes (TC) with telopodes contained secretory vesicles extended around the blood capillaries and contacted adjacent fibroblasts (Figure 8C). Many endocrine cells were determined in close contact with the blood vessels that were identified by the presence of mitochondria and electron-dense granules (Figure 8D).

## 4. Discussion

The ovaries of zebrafish show oocytes at all phases of development and degeneration (atresia). This also occurs in cyprinids, tilapia, and other teleosts due to multiple spawning, which is indicated by the occurrence of asynchronous oocyte development. Ovulation and spawning can happen almost simultaneously in some fish, while, in others, such as the Salmonidae family, the ovulated eggs are kept in the peritoneal cavity and spawning occurs later [22].

The early oocyte differentiation and ovarian development in the zebrafish have not yet been fully utilized. In the present study, based on morphological and structural criteria, the ovaries of the zebrafish *D. rerio* showed oocytes at all phases of development and degeneration (atresia). Six stages of oogenesis were identified as follows: oogonia, early oocytes, late oocytes, vacuolated follicles, the yolk globule stage (vitellogenesis), and mature follicles. However, the stages of oocyte development in the zebrafish are divided into five stages: the primary growth stage (I), which includes two morphological forms, IA and IB; the cortical alveolus stage (II); vitellogenesis (III); oocyte maturation (IV); and mature eggs (V) [1]. Stages IA and IB are morphologically analogous to the oogonia, and early oocytes described in the current study. On the other hand, the primary growth, cortical alveolus, vitellogenic, and mature oocyte stages were identified as the four stages of zebrafish (*D. rerio*) oocyte development [23].

The yolk globule stage is considered the most crucial stage of oocyte development as vitellogenesis occurs during this stage, resulting in the comprehensive growth of oocytes mainly by the rapid integration of high levels of exogenous vitellogenin produced from the liver [24]. Some authors have reported that the growing ovarian follicles produce steroid hormones (estradiol). Through blood arteries that supply the theca cell layer, this steroid leaves the follicle and travels to the liver, where it induces the manufacturing of vitellogenin. Vitellogenin is transported through the bloodstream to the ovary, where it is absorbed by the egg and deposited as yolk protein, which is used as a building material and an energy source after fertilization [25]. The present study showed that the yolk globules consisted mainly of lipoprotein with a small proportion of carbohydrates and displayed strong reactivity to the Sirus red stain. Besides being the major nutritional source, yolk proteins are useful for monitoring the environment [26].

A previous study has identified the initial phases of oocyte differentiation and ovarian development in zebrafish and established precise staging standards [27]. In this study, the authors discovered new cytoskeletal structures in oocytes and determined the role that the specialized cellular architecture plays in differentiation. In the present work, TEM allowed the identification of several cellular structures, including stem cells and telocytes, which are supposed to play potential roles in ovarian physiology, differentiation, and development.

Atresia tends to occur more commonly in vitellogenic follicles than in previtellogenic ones. On the other hand, some authors have mentioned that a few oocytes collapse due to failure to reach maturation; this may be due to environmental factors (photoperiod and water temperature) or feeding conditions [28]. Oocyte degeneration happens if metabolic or endocrine problems manifest during vitellogenesis. At the beginning of atresia, the zona radiata became convoluted and start to disintegrate. The corpus atreticum is a compact, well-vascularized tissue formed by the proliferation and hypertrophy of thecal and follicular granulosa cells. By breaking down the zona radiata, these active granulosa cells invade the oocyte and actively phagocytose the yolky contents [29]. Therefore, degenerated yolky oocytes are a sign of recent spawning.

Myostatin (MSTN) is a member of the transforming growth factor β (TGF-β) superfamily and was first demonstrated as a negative growth regulator of the skeletal muscles [30]. In recent years, the functions of MSTN in other tissues rather than the musculoskeletal system have attracted researchers’ interest. MSTM is widely expressed in several tissues of fish, including the pseudobranch [31], liver [32], and ependymal cells [33].

The present study showed the positive immunoreactivity of myostatin in both granulosa and theca cells. Similar findings were observed in zebrafish and sea bream, indicating its potential function in female reproduction [34]. It is interesting to note that in brook trout, myostatin mRNA levels were found to be highly elevated during ovulation [35]. Numerous studies have confirmed that MSTN is essential for reproduction as it controls oocyte maturation, granule cell proliferation, ovarian steroidogenesis, and follicular development. [36]. Wang et al. (2022) added that any alteration in MSTN or its receptors may influence ovarian function [37].

During gonad development, MSTN may have a function as a growth regulator [38]. The two key follicular development stages that are necessary for oocyte maturation and ovulation are the proliferation and terminal differentiation of granulosa cells [39]. Previous studies have elucidated that MSTN enhances the expression of connective tissue growth factor (CTGF) [40]. Theca cell recruitment and follicular development are two of the many ovarian processes that CTGF regulates [40].

In teleosts like zebrafish (*D. rerio*), medaka *(Oryzias latipes),* fugu (*Takifugu rubripes*), and rice field eel (*Monopterus albus*), two distinct forms of the SOX9 gene (*SOX9a1* and *SOX9a2*) have been demonstrated [12,13,41]. In the ovary of medaka, SOX9 was expressed [42]. In zebrafish, SOX9b was expressed in the ovary, while SOX9a was localized to the testis, similar to mammals [13]. The two SOX9a genes were immunoreactive in the testis, ovary, and ova-testis of rice field eel [41]. Moreover, SOX9b was localized in the testis of the common carp [43]. These results indicate that the SOX9 expression in the gonads is diverse in teleosts. Most of these studies were conducted on fortnightly or daily or breeders like medaka, zebrafish, and tilapia during the early stages of gonadal development. The present study showed the strong expression of SOX9 in the previtellogenic follicles, which indicated their role in follicular development. Bhat et al. (2016) added that in walking catfish (*Clarias batrachus),* SOX9 was expressed at a high level before spawning, but it steadily dropped during the oviposition period and after oviposition [44].

On the other hand, the SOXB1 subfamily in fish contributes to sex differentiation, sex determination, the development of gonads, the formation of many tissues and organs, and early embryonic differentiation [45]. Additionally, it has been discovered that SOX9 is crucial for gonad seasonal variations [44]. In addition, the expression of SOX9 has been identified in the muscles, liver, gonads, brain, gills, fins, eyes, kidneys, and spleen of sturgeons [46,47,48]. These works suggest that SOX9 plays critical roles in physiological and developmental processes in fish. Moreover, the current results show that SOX9 is expressed in the ovarian stem (progenitor) cells, which indicates that the development of germ cells and oogenesis are regulated by SOX9. These results are supported by the findings of Salvaggio et al. (2016), who confirmed that in zebrafish, the SOX family is crucial for maintaining stem cells and promoting embryonic development [49].

The interstitial compartment of the ovary is primarily made up of fibroblasts, which are connective tissue components, as well as steroidogenic cells, blood vessels, and a significant amount of extracellular matrix. The current work indicated the distribution of telocytes in the ovarian stroma of zebrafish. Telocytes were also observed in the ovaries of redbelly tilapia [50] and *Cyprinus carpio* [51]. The EM findings revealed the occurrence of heterocellular contact with blood vessels, fibroblasts, and theca cells, which may be involved in the regulation of the regeneration process, hemostasis, and the remodeling of gonadal tissues during the spawning season [52]. Moreover, many secretory vesicles were observed in their telopodes between theca cells and around the blood vessels, indicating their role in intracellular communication. The present study showed the distribution of many immune cells in the ovarian stroma of zebrafish. The immune system plays a role in the control of gonad function [50].

The current study showed that macrophages are commonly distributed throughout the ovarian tissues and could be recognized by their distinct morphology and phagocytic features. These cells have multiple functions that are mainly implicated in immune reactions, including phagocytosis and the production of growth factors, chemokines, and cytokines, as well as the degradation of foreign antigens and tissue remodeling [53]. Furthermore, their distribution in the ovary of the zebrafish during the spawning season suggests that macrophages are involved in a variety of intra-ovarian processes, including folliculogenesis and tissue repair following atresia. The current observations reveal that the dendritic cells (DCs) are distributed in the ovaries of the zebrafish. DCs are antigen-presenting cells with a dendritic appearance, phagocytic capacity, and potent T-cell-stimulatory characteristics [54].

## 5. Conclusions

This study highlighted the cellular and microenvironmental changes of the ovary in the zebrafish during the spawning season. The ovaries of the zebrafish showed oocytes in all stages of follicular development and degeneration. The expression of SOX9 in the ooplasm of the primary and previtellogenic oocytes suggests their role in follicular development. Moreover, SOX9 expression in the ovarian stem (progenitor) cells indicates the regulatory role of SOX9 in the process of germ cell development and oogenesis. The immunoreactivity of myostatin in the granulosa and theca cells suggests their roles in oocyte maturation, granule cell proliferation, ovarian steroidogenesis, and follicular development. In addition, several cells were detected in the ovarian stroma, including mast cells, telocytes, lymphocytes, fibroblasts, endocrine cells, macrophages, adipocytes, dendritic cells, and steroidogenic (stromal) cells, indicating the multiple functions of the ovary. Accordingly, these findings suggest the ovary of the zebrafish as a model for the study of several aspects of ovarian development. However, further studies should be performed to identify the mechanisms through which SOX9 and myostatin regulate follicular development.

## Figures and Tables

**Figure 1 animals-13-03362-f001:**
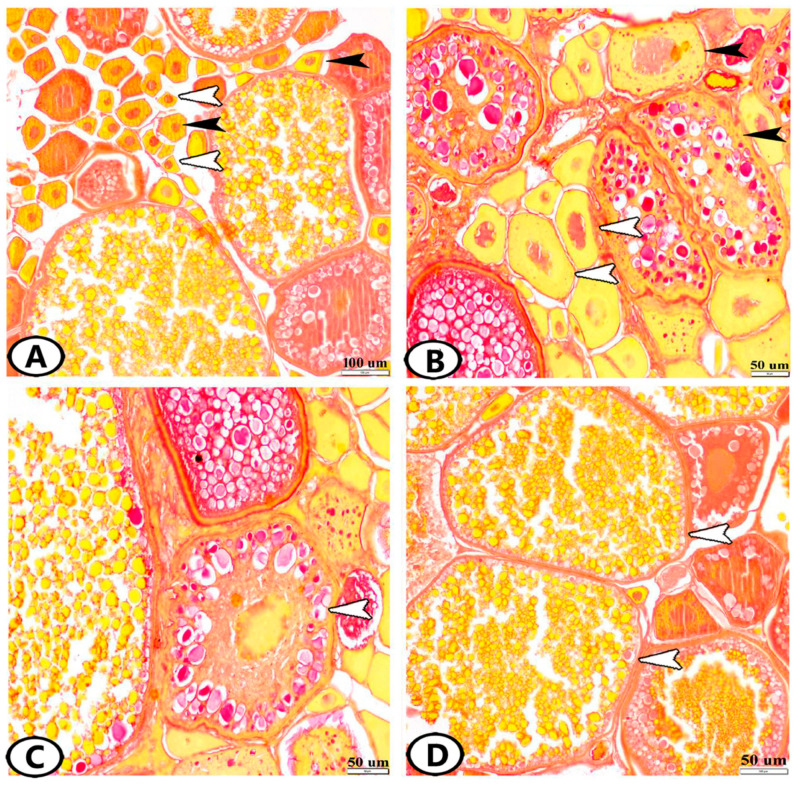
Histological structure of the ovary of zebrafish stained by Sirus red. (**A**) Oogonia (white arrowheads) and chromatin nucleolus stage (black arrowheads). (**B**) Perinucleolar stage (white arrowheads) and vacuolated follicles (black arrowheads). Note that the nucleoplasm attained red granules with Sirus red stain. (**C**) Yolk globule stage (white arrowhead). (**D**) Mature stage (white arrowheads).

**Figure 2 animals-13-03362-f002:**
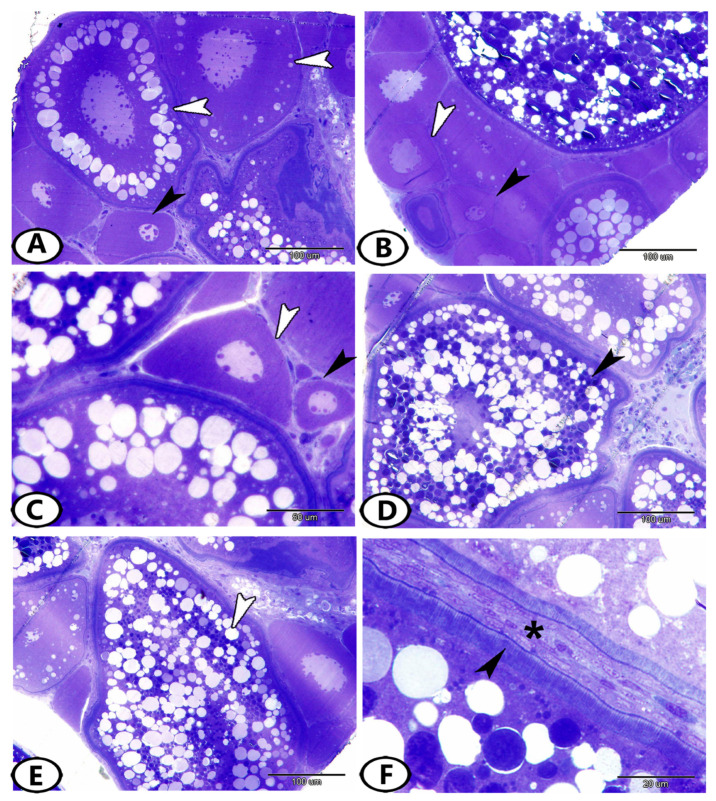
Semithin sections stained by toluidine blue show follicular development. (**A**) Early oocytes with many nucleoli (black arrowhead) and vacuolated follicles (white arrowheads). (**B**) Early oocytes (black arrowhead) and late oocytes (white arrowhead). (**C**) Late oocyte with flat follicular epithelium (white arrowhead). Note the early oocyte (black arrowhead). (**D**) Yolk globule stage (black arrowhead). (**E**,**F**) Mature follicle (white arrowhead). Note zona radiata (black arrowhead) and theca folliculi (asterisk).

**Figure 3 animals-13-03362-f003:**
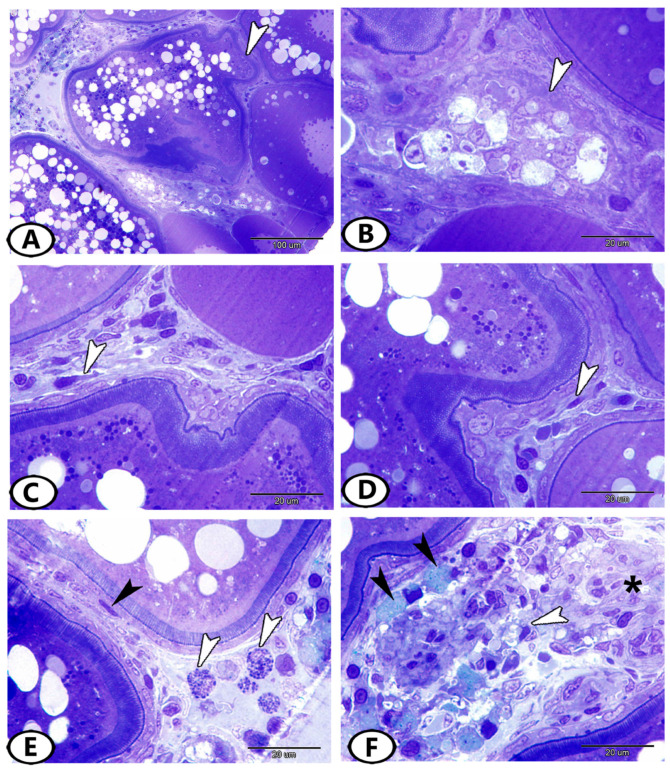
Semithin sections stained by toluidine blue show the atretic follicles and stroma. (**A**) Atretic follicle with irregular membrane (white arrowhead). (**B**) Advanced atretic follicle (white arrowhead). Note that the yolk is completely reabsorbed, while the zona radiata and follicular cells are completely digested. (**C**,**D**) The stroma contained fibroblasts (white arrowheads). (**E**) The stroma contained mast cells (white arrowheads) and telocytes (black arrowhead). (**F**) The stroma contained fat cells (black arrowheads), macrophages (white arrowhead), and stromal cells (asterisk).

**Figure 4 animals-13-03362-f004:**
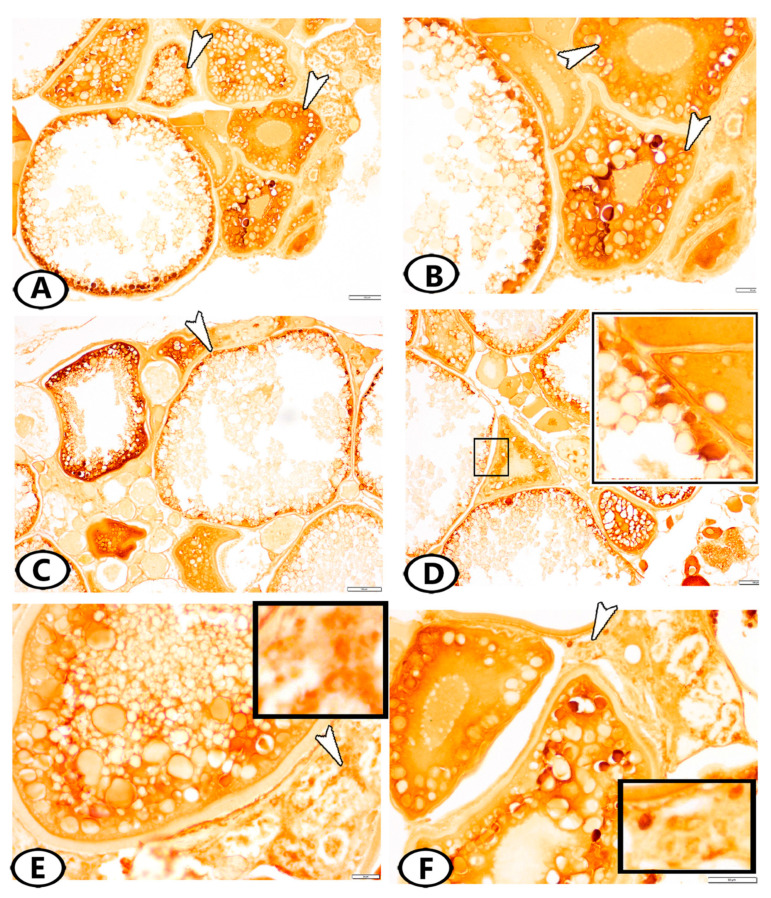
Immunohistochemistry of SOX9. (**A**,**B**) SOX9 was localized in the ooplasm of primary and pre-vitellogenic oocytes (white arrowheads). (**C**,**D**) SOX9 was expressed in the theca interna cell layer, which surrounded the mature follicles (white arrowhead, boxed areas). (**E**,**F**) Many stem cells (arrowheads, boxed areas) in the ovarian stroma expressed SOX9.

**Figure 5 animals-13-03362-f005:**
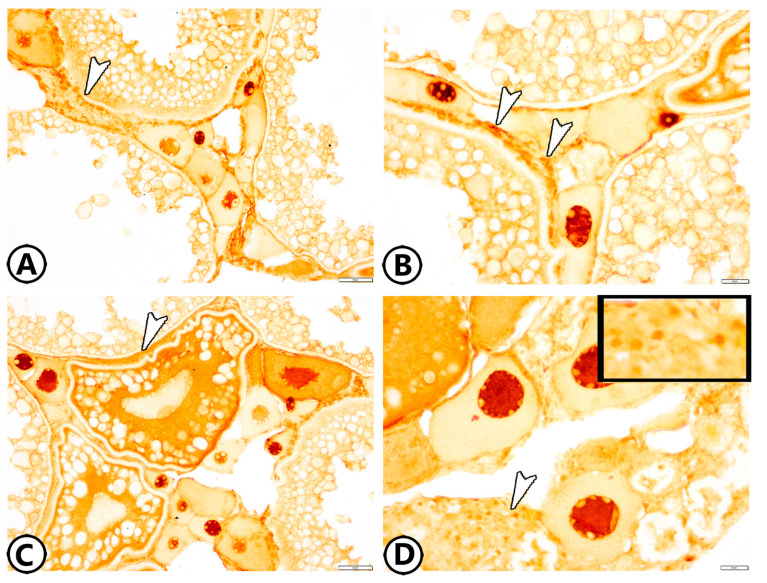
Immunohistochemistry of myostatin. (**A**–**C**) Myostatin was expressed in granulosa and theca cells of mature follicles (arrowheads). (**D**) Many stem cells (arrowhead, boxed area) in the ovarian stroma expressed myostatin.

**Figure 6 animals-13-03362-f006:**
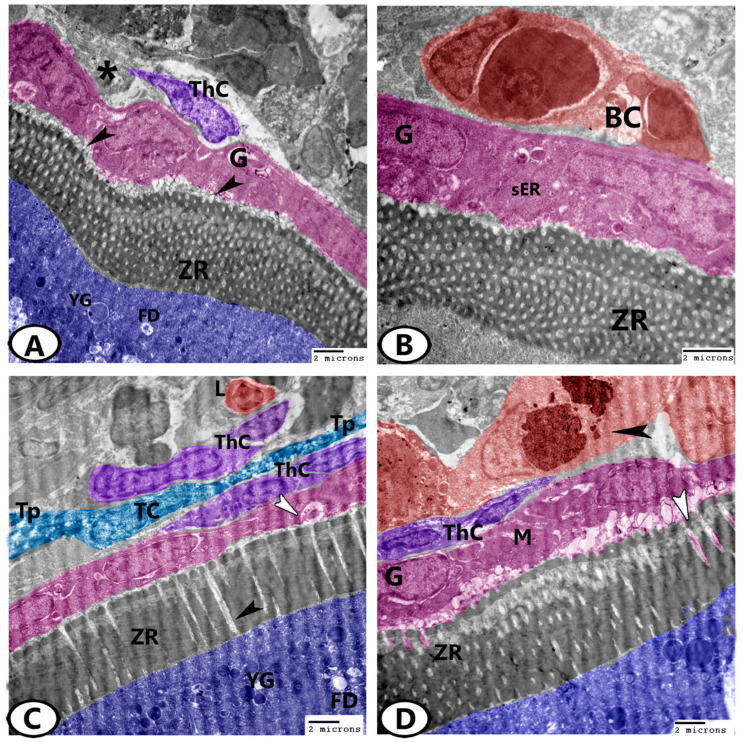
Digital colored TEM images of the mature ovarian follicles of zebrafish. (**A**) Fat droplets (FD) and yolk globules (YG) are found in the outer ooplasm of the mature follicles. Note the processes (arrowheads) of the granulosa cells (G, pink) that penetrated the zona radiata (ZR). Theca cells (ThC, violet) are embedded in the collagen network (asterisk). (**B**) Higher magnification shows blood capillaries (BC, red) in the thecal layer. Note the granulosa (G, pink) that contained sER and zona radiata (ZR). (**C**) The zona radiata (ZR) was traversed perpendicularly by pore canals (black arrowhead) containing processes from both oocytes (blue) and zona granulosa (G, pink, white arrowhead). The oocytes contained fat droplets (FD) and yolk globules (YG). Note the presence of telocytes (TC) and telopodes (Tp) between the thecal layers (ThC). Lymphocytes (L, red) were found in the ovarian stoma. (**D**) The granulosa layer (G, pink) contained many mitochondria (M). Note the processes (white arrowhead) of the granulosa layer that penetrated the zona radiata (ZR). Large macrophages (red, black arrowhead) could be seen neighboring the thecal layer (ThC, violet).

**Figure 7 animals-13-03362-f007:**
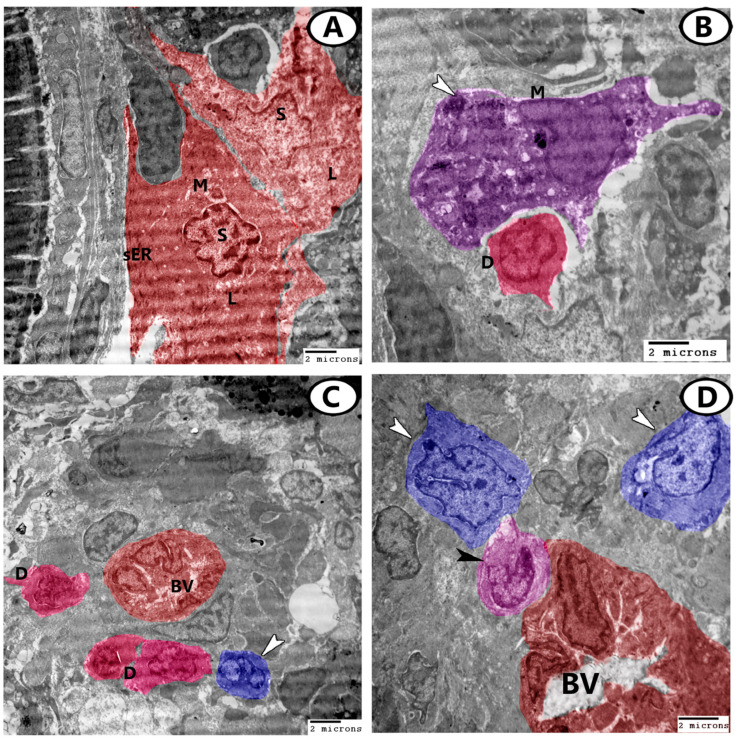
Digital colored TEM images of the ovarian stroma. (**A**) Two steroid interstitial stromal cells (S, red) contained sER, mitochondria (M), and many lipid droplets (L). (**B**) Dendritic cells (D, pink) could also be seen in the ovarian stroma in association with macrophages (M, violet) that contained heterogeneous materials (arrowhead). (**C**) Dendritic cells (D, pink) were distributed around the blood vessels (BV). Note the presence of stem cells (arrowhead, blue) with dividing nuclei. (**D**) Monocytes (black arrowhead) were in close contact with blood vessels (BV). Stem cells (white arrowhead, blue) were observed in the stroma.

**Figure 8 animals-13-03362-f008:**
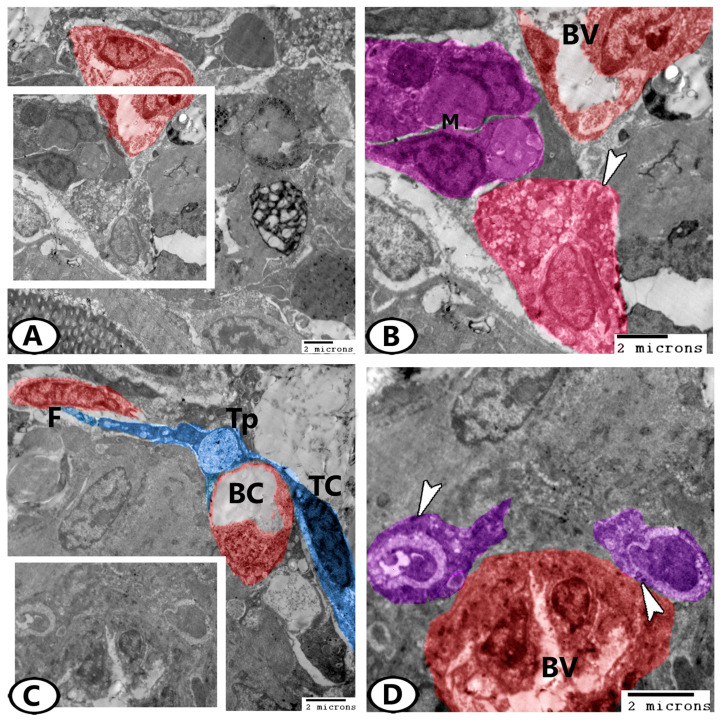
Digital colored TEM images of the ovarian stroma. (**A**,**B**) Low and higher magnifications show the fat cells (pink, arrowhead) and macrophages (violet, M) in association with blood vessels (BV, red). (**C**) Telocytes (TC, blue) with telopodes (Tp) contained secretory vesicles extended around the blood capillaries (BC) and contacted adjacent fibroblasts (F, red). (**D**) Higher magnification of the boxed area in (**C**) shows many endocrine cells (violet) with electron-dense granules (arrowheads) in close contact with the blood vessels (BV).

## Data Availability

The data presented in this study are available within the article.

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
