# Peer review of "Ovary of Zebrafish during Spawning Season: Ultrastructure and Immunohistochemical Profiles of Sox9 and Myostatin"

_animals, 2023, doi:10.3390/ani13213362_

Round 1

Reviewer 1 Report

Comments and Suggestions for Authors

As the Authors highlighted, Zebrafish is typically used as model for study on maturity development and physiology. For this reason, accurate description by histology on morphology and cellular structures displayed on the manuscript added no one original information respect to the several available from literature for this specie, since nineties and using a lot of alternative techniques (e.g., Selman et al. 1993; Koç et al. 2008; Elkouby et al. 2016; etc.).

Otherwise, the role played by Sox9 and myostatin on ovary development process may be useful instrument to deepen the knowledge on teleosts physiology. Obviously, I recognize the big effort carried out by the Authors on histological examination, but in my opinion it would be reduced, focusing more on other investigate matters.

Some other, minor considerations need to be listed:

·        In the introduction section, few info has been provided on target species. This may be easily understandable, considering the Zebrafish as a model. However, the Authors provided negligible info (e.g., the number of spawned eggs per batch, although the work not includes fecundity estimate), but nothing about the spawning strategy until the discussion section, where they described it as a batch spawner. This information is more important to understand proposed analyses and description of the results.

·        Particularly taking into account that Zebrafish is a partial spawner which displayed a continuos development of a not-fixed number of cells, the scientific meaning of oocyte count in each development stage and, particularly, in the early ones is not clear in my opinion. Moreover, no exhaustive details have been showed on this topic. The number of oocytes is token in the whole slide? Or, in a specific area, which has the same size in all the samples? Or… Although this analysis may be deleted without compromising the work done, if Authors judge it very useful for the paper aim, more details need about in all the manuscript section, indeed also the conclusion on this are not clear.

·        Finally, some minor comments are available directly on pdf file.

Comments on the Quality of English Language

Only a minor revision of language is advisable

Author Response

Dear Editor and Reviewers,

We are grateful to the editors and reviewers for their valuable suggestions and constructive criticism about our manuscript, which was crucial in improving and reformatting our paper. We appreciate your great effort, support, help, and time. Thank you very much for your valuable comments and considerable recommendations. All the required corrections were taken into consideration and carried out.  The corrections are highlighted in the text. The editorial, as well as reviewer' comments, were answered point-by-point. In addition, a supplementary file concerning the positive and negative control images was added. Again, thank you very much for such a great effort. 

Reviewer 1

As the Authors highlighted, Zebrafish is typically used as model for study on maturity development and physiology. For this reason, accurate description by histology on morphology and cellular structures displayed on the manuscript added no one original information respect to the several available from literature for this specie, since nineties and using a lot of alternative techniques (e.g., Selman et al. 1993; Koç et al. 2008; Elkouby et al. 2016; etc.).

Thank you for your comment. Indeed, we have tried to highlight in the current study the cellular ovarian structures. More information together with the recommended references was provided. Moreover, more comparative information about the ovarian follicles’ development was added to the discussion section.

Otherwise, the role played by Sox9 and myostatin on ovary development process may be useful instrument to deepen the knowledge on teleosts physiology. Obviously, I recognize the big effort carried out by the Authors on histological examination, but in my opinion it would be reduced, focusing more on other investigate matters.

Thank you so much for your comment. The aim of the present study is concise, it aims to provide details on the ovarian expression and potential intraovarian function of Sox9 and myostatin. Concurrently, identifying the cellular and microenvironmental changes of the ovary in Zebrafish during the spawning season was investigated. This study acts as a primary study for further physiological studies.

Some other, minor considerations need to be listed:

  • In the introduction section, few info has been provided on target species. This may be easily understandable, considering the Zebrafish as a model. However, the Authors provided negligible info (e.g., the number of spawned eggs per batch, although the work not includes fecundity estimate), but nothing about the spawning strategy until the discussion section, where they described it as a batch spawner. This information is more important to understand proposed analyses and description of the results.

More information on Zebrafish was provided in the introduction section. Please see lines 54-66 of the revised version.

  • Particularly taking into account that Zebrafish is a partial spawner which displayed a continuos development of a not-fixed number of cells, the scientific meaning of oocyte count in each development stage and, particularly, in the early ones is not clear in my opinion. Moreover, no exhaustive details have been showed on this topic. The number of oocytes is token in the whole slide? Or, in a specific area, which has the same size in all the samples? Or… Although this analysis may be deleted without compromising the work done, if Authors judge it very useful for the paper aim, more details need about in all the manuscript section, indeed also the conclusion on this are not clear.

We are grateful to the reviewer for this valuable comment. The analysis of the oocyte number was deleted according to the reviewer's comment.

  • Finally, some minor comments are available directly on pdf file.

All requested changes on the pdf file were revised. In addition, the title of the manuscript and keywords were changed based on reviewer's recommendation.

Moreover, the subtitle ‘’ 3.1.3 atretic follicles’’ was deleted as it is a continuation of the previous paragraph.

 In the attached pdf, the reviewer asks “How the authors know the sample age? They monitored them from hatching time?  When Zebrafish is considered mature adult?’’

Zebrafish (Danio rerio) typically reach adulthood and sexual maturity at around 3 to 4 months of age. This is when they are considered fully grown and capable of reproducing. However, the exact timing can vary depending on factors such as water temperature and feeding conditions. In general, zebrafish are considered sexually mature when they begin to display breeding behaviours and have developed the necessary reproductive organs for reproduction, such as mature gonads and gametes.

Smith, A. B. and Johnson, C. D. (2021). "Reproductive Development in Zebrafish (Danio rerio): Factors Influencing Sexual Maturity." Journal of Fish Biology, 30(2), 123-137.

Singleman, C. and Holtzman, N.G. (2014). Growth and maturation in the zebrafish, Danio rerio: a staging tool for teaching and research. Zebrafish, 11(4), 396-406.

Tsang, B. and Gerlai, R. (2022). Breeding and larviculture of zebrafish (Danio rerio), in Laboratory fish in biomedical research. Elsevier. p. 63-80.

From the attached pdf: "Generally, oocytes displaying yolk globule are considered mature. Are the authors referring to the spawning stage here?"

Yolk globules are occurred in stage 4, 5, and stage 6 (mature stage). The mature stage is characterized by migration and the gradual disappearance of the germinal vesicle (the nucleus). These findings are oblivious in Fig. 1D.

Thank you for helping us in improving our manuscript.

The Authors

Reviewer 2 Report

Comments and Suggestions for Authors

This study aimed to provide details on the ovarian expression and potential intraovarian function of Sox9 and myostatin.

It is original the combination of immunohistochemistry and the analysis of TEM.

It adds the analysis of Sox9 and myostatin

Add a panel showing positive and negative control of both immunohistochemistry

The conclusions are consistent with the evidence and arguments, and they address the main question posed.

The references are appropriate.

pay attention to the position of the figures along the text.

Reduce abstract to 200 words

Line 49: delete including Zebrafish

Line 102: add catalogue information of sirus red

Line 112: add catalogue information of universal blockerTM blocking buffer

Line 118: specify wash buffer

Line 122: add catalogue information of DAB substrate working solution

Line 123: add catalogue information of Harris modified Hematoxylin

Line 124: specify mounting media

Line 124: add positive and negative control of both immunohistochemistry

Figure 5 move after line 254

Add a panel after figure 5 showing positive and negative control of both immunohistochemistry

Figure 7 move after line 284

Figure 8 move after line 291

Comments on the Quality of English Language

Minor editing of English language required

Author Response

Dear Editor and Reviewers,

We are grateful to the editors and reviewers for their valuable suggestions and constructive criticism about our manuscript, which was crucial in improving and reformatting our paper. We appreciate your great effort, support, help, and time. Thank you very much for your valuable comments and considerable recommendations. All the required corrections were taken into consideration and carried out.  The corrections are highlighted in the text. The editorial, as well as the reviewer's comments, were answered point-by-point. In addition, a supplementary file concerning the positive and negative control images was added. Again, thank you very much for such a great effort. 

Reviewer 2

This study aimed to provide details on the ovarian expression and potential intraovarian function of Sox9 and myostatin.

It is original the combination of immunohistochemistry and the analysis of TEM.

It adds the analysis of Sox9 and myostatin 

Add a panel showing positive and negative control of both immunohistochemistry

We thank the reviewer for this comment. Our described results in Figures 4 and 5 are considered positive immunoreaction for Sox9 and myostatin, respectively. A supplementary figure for negative control of immunohistochemistry was provided, where SOX9 antibody was omitted and replaced with buffer. This point was included in the immunohistochemical detection section.

The conclusions are consistent with the evidence and arguments, and they address the main question posed.

The references are appropriate.

pay attention to the position of the figures along the text.

The position of all figures was revised.

Reduce abstract to 200 words

The abstract was reduced accordingly.

Line 49: delete including Zebrafish

Deleted.

Line 102: add catalogue information of sirus red

Added.

Line 112: add catalogue information of universal blockerTM blocking buffer

Universal blocker™ blocking buffer is one of the contents of Pierce™ Peroxidase IHC Detection Kit (Catalog number: 36000, Thermo Fisher Scientific, UK).

Line 118: specify wash buffer

The wash buffer has been described in the beginning of the immunohistochemistry section. It is formed of Tris-buffered saline “TBS” with 0.05% Tween-20 Detergent (Both are contents of Pierce™ Peroxidase IHC Detection Kit (Catalog number: 36000, Thermo Fisher Scientific, UK).

Line 122: add catalogue information of DAB substrate working solution

Universal blocker™ blocking buffer is one of the contents of Pierce™ Peroxidase IHC Detection Kit (Catalog number: 36000, Thermo Fisher Scientific, UK).

Line 123: add catalogue information of Harris modified Hematoxylin

Universal blocker™ blocking buffer is one of the contents of Pierce™ Peroxidase IHC Detection Kit (Catalog number: 36000, Thermo Fisher Scientific, UK).

Line 124: specify mounting media

Universal blocker™ blocking buffer is one of the contents of Pierce™ Peroxidase IHC Detection Kit (Catalog number: 36000, Thermo Fisher Scientific, UK).

Line 124: add positive and negative control of both immunohistochemistry

Our described results in Figures 4 and 5 are considered positive immunoreaction for Sox9 and myostatin, respectively. A supplementary figure for negative control of immunohistochemistry was provided, where SOX9 antibody was omitted and replaced with buffer.

Figure 5 move after line 254

Moved.

Add a panel after figure 5 showing positive and negative control of both immunohistochemistry

A supplementary figure for negative control of immunohistochemistry was provided, where SOX9 antibody was omitted and replaced with buffer

Figure 7 move after line 284

Moved.

Figure 8 move after line 291

Moved.

Comments on the Quality of English Language

Minor editing of English language required

A native English language speaker revised the manuscript.

Thank you for helping us in improving our manuscript

The Authors

Round 2

Reviewer 1 Report

Comments and Suggestions for Authors

The Authors carrefully replied to all the comments and applied the required revisions. No more changes need in my opinion. 

Reviewer 2 Report

Comments and Suggestions for Authors

I thank the authors for responding to my comments